# “Forced Transformation” or “Regulation Capture”—Research on the Interactive Mechanism between Environmental Regulation and Green Transformation of Dairy Farming Subject Production

**DOI:** 10.3390/ijerph191912982

**Published:** 2022-10-10

**Authors:** Jiabin Xu, Tianyi Wang, Jingjing Wang, Cuixia Li, Limei Zhao

**Affiliations:** 1College of Economics and Management, Northeast Agricultural University, Harbin 150030, China; 2Institute of Rural Development, Sichuan Academy of Social Sciences, Chengdu 610071, China

**Keywords:** environmental regulation, green reputation, dairy farming subject, green transformation of production

## Abstract

Under the situation of an increasing resource and environment shortage, the green transformation of dairy farming subject production driven by environmental regulation is the concentrated embodiment of a “promising government” to solve the problem of breeding environmental pollution. Due to the shortcomings of environmental regulation itself and the undefined connotation of the green transformation of dairy farming subject production, the interactive relationship between the two remains unclear at present. Based on defining the concept of green transformation of dairy farming subject production, this paper aims to analyze the interactive mechanism between the environmental regulation and green transformation of dairy farming main production, build a dynamic game model between the environmental regulators and dairy farming subject, and introduce the constraints and benefits of a reputation mechanism on the behavior in the model to explore whether environmental regulation can drive the green transformation of dairy farming subject production. The results showed that the green transformation of dairy farming subject production followed the “subject substitution view” and emphasized “source reduction, process control and terminal treatment”. Strictly designed environmental regulations could effectively drive the green transformation of dairy farming subject production, but it was inevitable that the environmental regulators were vulnerable to the rent-seeking behavior of dairy farming subjects, which was “regulation capture”. The introduction of the reputation mechanism has greatly improved the rent-seeking behavior of dairy farming subjects and the probability that environmental regulators have “regulation capture”, indirectly forcing dairy farming subjects to participate in the green transformation of production. The greater the punishment for dairy farming subjects who do not participate in the green transformation of production was, the more they can be forced to participate in the green transformation of production. At the same time, it also reduces the risk of damage to the credibility of the government. Based on the studies above, this paper also further discussed the shortcomings of environmental regulation itself, including the “re exit and light implementation” of the environmental regulation policy, “decentralization and light inspection” of the environmental regulation subject, “result and light process” of the environmental regulation mode, and “formal regulation and light informal regulation” of the environmental regulation form, which provides a scientific reference for the formulation of the environmental regulation policy of livestock and poultry breeding in the future. Compared with previous studies, this paper is innovative in two aspects: first, it defines the conceptual connotation of a green transformation of dairy farming subject production, and second, it systematically discusses the interaction mechanism between the environmental regulation and green transformation of dairy farming subject production. This paper provides a scientific reference for the development of future environmental regulation policies for livestock and poultry farming.

## 1. Introduction

According to the Second National Pollution Source Census Bulletin jointly released by the Ministry of Ecology and Environment, the National Bureau of Statistics, and the Ministry of Agriculture and Rural Affairs in June 2020, in 2017, the emissions of chemical oxygen demand, ammonia nitrogen, total nitrogen, and total phosphorus, which made up the water pollution emissions from the livestock and poultry breeding industry, were 10.5 million tons, 11.9 million tons, 59.6 million tons, and 12.0 million tons, respectively, accounting for the agricultural sources of water pollutant emissions. The livestock and poultry farming industry has become an important source of pollution in the agricultural industry. As an important part of the livestock industry, dairy cattle are the third largest producer of pollution after pigs and beef cattle. From the perspective of pollutant production, the annual manure production of dairy farming in 2020 was as high as 133,243,000 tons, accounting for 48.2% of the total manure production of animal husbandry, which is a veritable livestock farming “major polluter”. From the perspective of carbon constraint, the livestock sector accounts for about 15% of total global carbon emissions, and 61% of carbon emissions in the livestock sector come from raising cattle [1]. In 2018, GHG emissions from the livestock sector in China were about 0.3 megaton CO_2_-eq, of which dairy farming accounted for about 13.3% [2]. Related studies have also shown that the annual ecological damage caused by greenhouse gases from ruminants such as beef cattle, dairy cattle, and sheep is 29% of the total annual ruminant production [3]. It can be seen that in terms of solving environmental pollution in dairy farming and promoting the green transformation of dairy farming, it is not only an essential requirement to have the high-quality development of dairy farming, but also to have an important performance of a “responsible power” under the constraints of the “emission peak and carbon neutrality goals”.

Green transformation is an inherent requirement for promoting high-quality economic development [4], and “emission reduction” as well as “efficiency enhancement” should be considered. The green transformation of the dairy farming industry and the overall green transformation of economic and social development must happen simultaneously. Combined with the dynamic characteristics of the farming system, the final aim is the continuous rationalization of resource allocation, pollution emission reduction, production efficiency, and environmental awareness of the farming subject. However, due to the public properties of the ecological environment itself and the negative externalities of environmental pollution, it is difficult for the market mechanism to effectively solve the problem of the environmental pollution of livestock and poultry breeding, and the government needs to “give a hand” through environmental regulation [5]. Government policies, regulations, and even environmental inspections are essential environmental regulations [6]. The practice shows that since 2000, China has gradually built an environmental regulation policy system that includes compulsory orders, economic incentives, and persuasion, together with education, to control pollution from livestock and poultry farming. However, on the one hand, most of these studies have focused on the environmental problems of pig farming and ignored dairy farming, which also is a “major polluter”. On the other hand, even if the farming environment regulation policy system gradually improves, the farming environment pollution problem still has not been effectively curbed. Therefore, the following questions are put forward: Is there a problem with environmental regulation itself? Does the introduction of environmental regulation provide incentives for dairy farming subjects to engage in a green transformation of production, or are environmental regulators captured by special interests? How should the regulatory role of environmental regulators be defined? These will be the scientific questions tackled in this paper. 

The papers directly related to this study focus on these aspects: First of all, the connotation of environmental regulation is studied. Regulation, also known as government regulation or control, is a mechanism designed by the government to restrain and restrict certain economic activities of regulated subjects for the overall welfare of society and achieve the redistribution of social resources through administrative intervention. Walter et al. firstly focused on environmental regulation and put forward the “pollution paradise hypothesis” [7]. Albrizio et al. pointed out that highly polluting and inefficient firms would lose their comparative advantage in developed regions with high environmental quality requirements and would generally move to less developed regions where environmental regulations are more lenient [8]. Since then, environmental regulation has become a hot topic in academic circles with the aim to solve environmental pollution problems [9,10]. Environmental regulation generally includes formal and informal environmental regulations according to the classification [11]. Formal environmental regulations include command and market incentives and informal environmental regulations include public participation and voluntarism. In terms of environmental regulation of livestock and poultry farming, although there is no clear connotation, common sense refers to the pursuit of economic maximization by farming subjects at the expense of ecological optimization, forcing the government to take the necessary regulation or control means to restrain, stimulate, or guide the way to supervise farming subjects to implement the resource utilization of farming waste and reduce the damage to the environment [12,13].

Secondly, the resource utilization of livestock and poultry breeding waste is studied. The livestock breeding process will unavoidably produce many undesired outputs, including manure discharged by livestock, food residues, breeding sewage, barn bedding, scattered feathers, and others [14]. Therefore, waste resource utilization is essential for alleviating the environmental pollution of livestock and poultry breeding in the future [15]. It is necessary to transform livestock and poultry farming waste into energy products and inputs for the planting industry through technical or management measures to achieve a healthy cycle between the planting industry and farming industry. Extensive academic research on the factors influencing the willingness of farming subjects’ waste resource utilization has led to the following three influential characteristics: One of them is the production and operation characteristics of the farming subject, including the scale of livestock and poultry farming [16], the business model [17], organizational relationships [18], etc. The second is the psychological characteristics of the farming subject, including environmental risk perception [19], resource utilization perception [20], revenue risk prediction [21], etc. The third is the external environmental characteristics of the farming subject, including social capital, the degree of agricultural marketization, public relationships, technology accessibility, and others [22,23].

Finally, the influence of environmental regulations on the decisions regarding waste resource utilization of farming subjects is studied. Based on the division of the Organization for Economic Cooperation and Development, the relevant environmental regulations are divided into two types, namely, the licensing system and subsidy system [24]. At present, the government usually implements subsidy policies to encourage farming subject entities to utilize waste resources [25]. Farming subjects are not only the targets of environmental regulation interventions, but also the implementers and beneficiaries of waste resource utilization [26]. In the process of enforcing regulations, the government grants subsidies to farming subjects to achieve the goal of “emission reduction”, which raises the level of environmental investment of farming subjects, reduces the influence from the external economy or external diseconomy, and further stimulates farming subjects to participate in reducing pollution [27]. After exploring the conclusion that environmental regulations can effectively drive the waste resource utilization of farming subjects, scholars have further studied the impact of different types of environmental regulations on the waste resource utilization decisions of farming subjects. Xu et al. pointed out that in rural areas where constraint-based environmental regulations are ineffective, guidance-based environmental regulations may have significant effects on promoting farming subjects to consciously practice pro-environmental behaviors [28]. Li et al. found that the hybrid environmental regulation (constraint + incentive) is more helpful to achieve the resourcefulness of waste by farming subjects [29]. Li et al. pointed out that some people argued that constrained environmental regulations are more effective than incentive-based environmental regulations in stimulating waste resource utilization decisions of farming subjects [26]. Zhu et al. concluded that guidance, constraint, and incentive environmental regulations have a significant impact on the decisions made around manure resource utilization by large-scale farming subjects, but the guidance environmental regulation plays the largest role [30]. At the same time, other scholars have included environmental regulations as a moderating variable in the analytical framework to conduct research [31,32].

The research results above provide useful empirical references for this study, but there is also space for further exploration: First of all, the resource utilization of livestock and poultry farming waste is not the same as the green transformation of production; resource utilization emphasizes the “terminal treatment” and ignores the “source reduction” and “process control”, which are necessary to define the connotation of green transformation of dairy farming subject production. Secondly, studies have been conducted on the impact of environmental regulations on the decision making around waste resource utilization by farming subjects, mostly based on empirical evidence, which has significant practical guidance. However, the regulatory role of the subject of environmental regulation (i.e., the environmental regulator), has not been clarified, and a deeper exploration of this issue can effectively explain “why ”there is still so much pollution“ although so many environmental regulation policies are made”. Thirdly, the research has not yet addressed the issues of “forced transformation” and “regulation capture” of livestock and poultry farming. The quality and safety of dairy products need to be strictly controlled. Under the support of national policies, the current proportion of large-scale dairy farms that produce milk has been as high as 61.4% [33], and most of the subjects have been qualified for the enterprise. They are highly susceptible to rent-seeking behavior and regulatory capture, and thus, an in-depth study of this issue is also useful to explore the effects of environmental regulation.

Because of this, this paper examines the issue of environmental regulation and the green transformation of dairy farming subject production, explores whether they are “forced transformation” or “regulation capture”, and innovatively introduces the reputation mechanism, which takes the constraints and expected benefits of reputation for both the government and the dairy farming subject into account, changing the general phenomenon of “governance over prevention and control” in previous studies. The concrete structure of the paper is as follows: the second section is a theoretical analysis; the third section is the research design; the fourth section is a model solution and result analysis; the fifth section is a further discussion; and the sixth section is the conclusion. 

Compared to previous studies, the possible marginal contribution of this paper is reflected in two aspects: First of all, the concept of green transformation of the production of dairy farming subjects is clearly defined and combined with established theory and field experience, the economic ideas of “subject substitution”, “emission reduction”, and “efficiency enhancement” are introduced in this paper, and the evaluation criteria of green transformation of dairy farming subjects are given. Secondly, the interaction mechanism between the environmental regulation and green transformation of the production of dairy farming subjects is systematically demonstrated. This paper constructed a dynamic game model between the government and dairy farming subjects based on game theory, and the constraints and benefits of the reputation mechanism on the behavior of both are introduced into the model to explore whether environmental regulation can drive the green transformation of dairy farming subject production.

## 2. Theoretical Analysis

### 2.1. Concept Definition: Green Transformation of Dairy Farming Subject Production

To clarify the theoretical content of the green transformation of dairy farming subjects, we also need to start from the essence of “green transformation”. The term “green economy” was first mentioned by the British environmental economist Pearce in his book, Blueprint for a Green Economy, in 1989. However, no specific explanation was given, making the connotation of the concept rather vague. As research progresses, it defines “green economy” as “sustainable growth through environmentally beneficial or non-confrontational economic behavior that together enhance economic and environmental benefits, achieving sustainable growth” [34]. This is highly consistent with the connotation of “green transformation” being explored by the current academic community. In addition, in the study of the nature of industry greening theory, Luo et al. proposed the “view of industry substitution” [35], which means that one industry can replace another to achieve industrial economic growth and ecological environmental protection [36]. This “industry substitution view” can be applied to the study of the green transformation of dairy farming subject production, but for the individual within an industry it should be the “view of subject substitution”. In this context, the theoretical essence of the green transformation of dairy farming subject production can be summarized as the process of replacing polluting dairy farming subjects with environmentally friendly dairy farming subjects within the industry, so that the industry as a whole can achieve a process of “emission reduction” as well as “efficiency enhancement”.

From the essence of the theory of the green transformation of dairy farming subjects, we can find that the green transformation of dairy farming subject production strongly supports the green transformation of dairy farming, and the emphasis is on “production”, “green”, and “transformation” on the basis of the “subject substitution view”. First, “production” emphasizes the process. “Production” is a process, i.e., the process of dairy farming, which can be divided into the source, process and end. The source requires “reduction”, i.e., a reduction in factor inputs that tend to produce ecological damage. The process requires “control”, i.e., additional non-hazardous treatment facilities for the timely treatment of discharged waste. The terminal requires “treatment”, that is, to make the full use of these “misplaced resources” so that they “turn waste into treasure”, such as through the production of organic fertilizer. Second, “green” highlights the standard. From the perspective of economics, the connotation of “green” is found to be that “green” emphasizes a kind of standard, which requires that each subject in the economic environment will not cause an ecological crisis and social collapse due to the blind pursuit of economic growth. While making the “economic pie”, it is necessary to build a “green background”. As far as dairy farming is concerned, a large quantity of pollutants will inevitably be emitted in the process of breeding, but these pollutants, if discharged recklessly, are “misplaced resources”; thus, the standard for “green” dairy farming should be to spare no effort to “reduce”, “control” as far as possible, and make every effort to “turn waste into treasure”. Third, “transformation” emphasizes dynamic results. Based on the process of “production” and standards of “green”, “transformation” focuses on “turning” as a dynamic status. The final result is to replace the polluting dairy farming subject with an environmentally friendly dairy farming subject to meet environmental requirements, in line with the progressive law of economic growth.

The content described above is the detailed connotation of the green transformation of dairy farming subject production, but “transformation” is a process. This process will also inevitably produce certain results, namely, “emission reduction” and “efficiency enhancement”. For dairy farming, the connotation of “emission reduction” is certainly easier to understand, emphasizing the reduction in non-point source pollution emissions and carbon emissions, where the reduction of non-point source pollution emissions refers to the reduction in organic matter such as chemical oxygen demand, nitrogen, phosphorus, zinc, and copper in dairy manure emissions in excess of the standard. Carbon emission reduction refers to the reduction of carbon dioxide emissions in six major segments: cow gastrointestinal fermentation, the cow manure management system, cow feeding energy consumption, feed grain cultivation, feed grain transportation and processing, and raw milk processing. The connotation of “efficiency enhancement” needs to be more carefully analyzed. Traditional economic growth theory considers capital and labor as the important sources of economic growth, without considering the role of technological progress in economic development. The new economic growth theory makes up for the traditional economic growth theory’s neglect of technological progress by emphasizing the need to consider not only capital and labor, but also technological advancement as the core of economic growth. Therefore, combining the connotation of the new economic growth theory, the new development pattern of China’s economy, and the actual characteristics of the dairy farming industry, the “efficiency enhancement” is specifically divided into benefit increase and efficiency increase. Among them, the benefit increase emphasizes the reduction in the cost of dairy farming and the increase in the economic value created by dairy farming, responding to the value created by labor and capital. Additionally, efficiency increases can be judged using a combination of total factor productivities, responding to the role of technological advancement in economic growth. Based on the analysis above, in order to show the scientific connotation of the green transformation of dairy farming subject production more clearly, the evaluation index system is given here in this paper, which is shown in Table 1.

### 2.2. Analysis on the Interaction Mechanism between Environmental Regulation and Green Transformation of Dairy Farming Subject Production 

According to the neoclassical school of thought, environmental regulation increases the cost of institutional compliance through a “Cost effect” that increases the financial burden on firms to pay for environmental pollution practices and crowds out resources for green innovation [37,38]. However, the “Porter hypothesis” suggests that appropriate environmental regulations provide “compensating benefits” that exceed the costs of regulation by inducing firms to innovate with green technologies [39] while ensuring no significant reduction in corporate efficiency and that green innovations are applied to the production process, reducing reliance on original polluting production methods, and effectively avoiding environmental regulation costs [40]. According to the view mentioned above, environmental regulations act on the green transformation of dairy farming subject production through the “Porter effect” and “Cost effect”. From the perspective of a short-term and static analysis, under the environmental regulation constraint, dairy farming subjects need to pay certain additional costs to participate in the transition, which brings a certain crowding-out effect on productive investment, thus bringing about an increase in dairy farming costs and a decrease in output, which is not conducive to the participation of dairy farming subjects in the green transformation of production. From the perspective of a long-term and dynamic analysis, proper environmental regulations have a positive incentive effect on dairy farming subjects to improve production technology and optimize resource allocation, which is an important source to stimulate subjects to “compensate for technological innovation”. By increasing R&D and investment in clean production technology and upgrading breeding structures, the “environmentally friendly farming subject” will gradually replace the “polluting farming subject”, thus continuously promoting dairy farming subjects to participate in the green transformation of production (path ① and path ②).

Existing studies have analyzed and tested the influence of environmental regulations on the waste resource utilization decisions of farming subjects. However, the development and implementation of environmental regulation policies are highly complex and uncertain processes, which can give rise to new problems. The details are shown in Figure 1 for path ③ and path ④. These two paths are precisely the core topic of this paper, that is, whether the relationship between environmental regulation and the green transformation of dairy farming subject production is one of “forced transformation” or “regulation capture”, as will be described below.

In the case of path ③, the authority to enforce the law is given to the environmental protection regulator after the government has formulated the environmental regulation policy, which has the power to control the dairy farming subject in this case [41]. Assuming that the environmental regulator is an absolutely neutral subject at this time, it will certainly enforce the law impartially according to the content of the environmental regulation policy; manage, monitor, and punish the environmentally unfriendly behavior of dairy farming subjects by setting a reasonable regulatory program; and make the dairy farming subjects who participate in the green transformation of production on time and according to the rules gain reputation by introducing a reputation mechanism, at which time the “forced transformation” plays an obvious role, which is reflected in both external pressures and internal incentives for dairy farming subjects. For external pressures, the green transformation of production is a realistic demand of external stakeholders for polluting dairy farming subjects, forcing them to weigh the consequences caused by their pollution, influencing the way they respond to the environmental regulations of dairy farming subjects, and accelerating their enthusiasm and initiative to participate in the green transformation of production. For internal motivation, the realization of the green transformation of production is a comprehensive reflection of the social and economic benefits of the dairy farming subjects. Environmental regulators’ inspections prompt dairy farmers to reflect on the shortcomings of their own green production of transformation, effectively remedying the inherent weaknesses of its governance mechanisms and overcoming its inertia to change, thereby accelerating the green transformation of production while reaping reputational benefits [42].

However, an environmental regulator with enforcement powers is often not an absolutely neutral subject, which results from the fact that it is composed of a series of sub-sectoral subjects with different structures, motivations, interest stimuli, and operational levels. Therefore, the looseness of the organization of the environmental regulator brings opportunities for dairy farming subjects to take advantage of the situation and trigger rent-seeking behavior, as shown in path ④. According to statistics, with the continuous improvement of the level of scale, standardization, and intensification, the dairy farming subject gradually advanced from the nature of “field (household)” to the nature of “enterprise”. By 2020, the top 20 dairy farming enterprises in China built 398 farms of their own, with a total cow stock of 1.715 million per head and raw milk production reaching 9.423 million tons [33]. In order to avoid punishment by environmental regulators, dairy farming subjects will adopt rent-seeking behavior, making environmental regulators “regulation capture”, which they combine with the attempt to deceive the government through opportunism to achieve the purpose of not participating in the green transformation of production while not being punished. However, although the environmental regulators have gained rent-seeking benefits, they have damaged the government’s credibility, brought about reputational damage, weakened the role of the environmental regulation policy, and will surely be severely punished by the government [43].

The goal of the free market is a matter of efficiency to increase social welfare with efficiency gains, but without the regulation of adequate policies, market welfare maximization is incomplete. Therefore, from the perspective of economic development, the general lack of regulation will lead to economic stagnation. From the ecological evolutionary perspective, the lack of regulation will, in turn, lead to a serious waste of primary productive resources. However, excessive government intervention can breed corruption in rent-seeking [44]. Therefore, whether environmental regulation can better drive the green transformation of dairy farming subject production depends on whether the stimulus of “forced transformation” is greater than the motivation of “regulation capture”, which is what we need to verify in the following part of this paper based on the mechanism of interaction between the environmental regulation and green transformation of dairy farming subject production.

## 3. Methods

### 3.1. Theoretical Introduction

The policy concept of behavioral economics emphasizes the stimulation of social preferences for members of society, rather than relying exclusively on material rewards and punishments [45]. Behavioral economists do not deny the existence of rationality, but at the same time, behavioral economists are appealing to the importance of intuition and emotion. Behavioral economics offers the concept of “boosting” oriented policy design to effectively stimulate people’s intrinsic social preferences, which means that some behavioral interventions are imposed on members of society at a low cost to stimulate their intrinsic social preferences and change their behavior in the direction of improving social welfare. In his book, *Human Agency and Behavioral Economics: Nudging Fast and Slow*, Cass R. Sunstein focuses on the scientific implications of “boosting” in human behavioral decision-making processes [46]. He suggests that boosting considers both human motivation and control, and that through the design of some mechanisms, people can be motivated to induce certain behaviors that are consistent with specific goals, but without compromising their freedom of choice.

As described by the essential connotation of behavioral economics, the green transformation of dairy farming subject production is a behavioral decision that determines whether the dairy farming subject participates in the green transformation of production or not. According to behavioral economics, the essence of economic activities in any industry is the process of human economic behavior, and human behavior is often short-sighted to the “invisible” future, even if the returns may be higher, while also reluctant to take risks. For dairy farmers to participate in the green transformation of production, the base period investment may be higher than the expected benefit, and this “uneconomical” investment will seriously affect the behavioral decisions of dairy farming subjects. Under the market mechanism, whether it is the reduction of antibiotics at the source of dairy farming, the procurement of environmentally sound equipment in the process, or the end of resource-based equipment, all reflect the characteristics of high cost and high risk, often not favored by dairy farming subjects. In particular, for small and medium-sized farming enterprises (farms) limited to cost–benefit considerations, there is a long way to go to achieve the standard of green transformation of production. Environmental regulation at the institutional level and reputation mechanisms at the government and social levels are needed to regulate and guide the cognitive process of dairy farming subjects, intervene in their production decisions, and motivate them to change their behavior to collectively rationalize and achieve the goal of the green transformation of dairy farming.

The trade-off between the goal of the green transformation of dairy farming and the participation of dairy farming subjects in the green transformation of production is usually considered to be between social welfare and private costs. However, the questions are: How can we effectively weigh the public’s desire for an eco-friendly society against the economic costs to be borne by polluting farming entities to produce a green transformation? How should the impact of government environmental regulation on the green transformation of dairy farming subject production be perceived? Can environmental regulations have a binding effect on the non-production green transformation choices of dairy farmers? Has the introduction of a series of environmental regulation policies motivated dairy farming subjects to take effective and innovative initiatives to engage in the green transformation of production or is there indeed a persistence of pollution due to the capture of environmental regulators by special interest groups? If an effective incentive can be achieved, dairy farming subjects will have to implement a green transformation of production in accordance with the environmental regulators of the government, which means that environmental regulations achieve the goal of “forced transformation”. However, if effective incentives are not realized, the rent-seeking behavior of the dairy farming subject achieves its goal, and environmental regulators would obtain “regulation capture”; therefore, the question of how to make environmental regulation effective needs to be reconsidered.

Based on the analysis above, this paper will construct a dynamic game model between the enforcer of environmental regulation, the environmental regulator, and the enforced, the dairy farming subject, from the perspective of game theory. Among the actions involved by environmental regulators are inspections and non-inspections, “regulation capture” and no “regulation capture”. The dairy farming subject is involved in behaviors that include participating in the green transformation of production and not participating in it, or rent-seeking and not rent-seeking. This paper also introduces reputation mechanisms into the model to further enhance the understanding of the impact of environmental regulations on the green transformation of dairy farming subject production.

### 3.2. Research Hypothesis

In a general sense, the government has a principal–agent relationship with environmental regulators. The government is the principal and the environmental regulator is the system agent. The government has delegated its rights to environmental regulator authorities in the hope that under the constraints of strict environmental regulations, environmental regulators can enforce the law according to the rules and urge dairy farming subjects to participate in the green transformation of production in order to accelerate the goal of the green transformation of dairy farming. Therefore, the environmental regulator is the one who has the discretion to judge whether the dairy farming subject is involved in the green transformation of production, and will rely on that discretion to decide whether to enforce penalties on the dairy farming subject. For this reason, a series of theoretical and practical research hypotheses need to be made before the construction of the specific model in order to better analyze the interaction mechanism between environmental regulation and the green transformation of dairy farming subject production.

#### 3.2.1. Basic assumptions

**Hypothesis** **1:**
*Dairy farming subjects, as “social beings” and “rational economic beings”, must be aware of the negative ecological impacts of not participating in the green transformation of production. Therefore, even if they are not willing to participate in the green transformation of production, dairy farming subjects will disguise their behavioral motives and claim that they are willing and proactively involved in it.*


**Hypothesis** **2:**
*As the enforcing body of environmental regulation, the environmental regulators are responsible for inspecting the transformation behavior of dairy farming subjects and making decisions on whether to punish them based on the inspection results. Thus, environmental regulators have the ability to accurately identify and determine whether the dairy farming subject is involved in a green transformation of production.*


#### 3.2.2. Assumptions on “rent-seeking” and “regulation capture”

**Hypothesis** **3:**
*If dairy farming subjects can consciously implement environmental regulations and participate in the green transformation of production, they do not need to seek rent from environmental regulators, and they do not need to consider whether environmental regulators conduct strict inspections; therefore, there is no “regulation capture”.*


**Hypothesis** **4:**
*If the dairy farming subject is not rent-seeking and there is no “regulation capture” by the environmental regulator, then the environmental regulator will collect the normal fines from them.*


**Hypothesis** **5:**
*If the dairy farming subject is not rent-seeking and there is “regulation capture” by the environmental regulators, then environmental regulators will use illegal fines to collect fines from dairy farmers above the normal fine level.*


**Hypothesis** **6:**
*If the dairy farming subject is rent-seeking and there is no “regulation capture” by the environmental regulators, the dairy farming subject will pay a bribe to the environmental regulators if it does not participate in the green transformation of production, but the environmental regulators will impose a fine on the dairy farming subject for not participating in the green transformation of production, confiscating the bribe money and handing it over to the higher authority; then, the regulator will receive an additional reward from the higher authority.*


**Hypothesis** **7:**
*If the dairy farming subject is rent-seeking and there is “regulation capture” by the environmental regulators, then it need not worry about whether to participate in the green transformation of production, but simply pays bribes to environmental regulators who will use the bribes as departmental revenue, or which may be privately used by some regulators.*


#### 3.2.3. The underlying assumptions of introducing the reputation mechanism

**Hypothesis** **8:**
*The government’s reputation has two components: first of all, if the environmental regulators do not perform their duties and do not regulate and punish dairy farming subjects, their credibility among the public will be lost; secondly, if the environmental regulators have “regulation capture”, the government’s credibility will also be seriously lost.*


**Hypothesis** **9:**
*The reputation of the dairy farming subject likewise consists of two components: first of all, if the environmental regulator conducts a routine inspection and finds that it has not participated in the green production transition as expected, the dairy farming subject is liable for a portion of the loss of reputation; secondly, if environmental regulators routinely inspect dairy farming subjects and check that they are better engaged in the green transformation of production, they will gain some reputational benefits.*


### 3.3. Model Construction

Before conducting the construction of the dynamic game model between environmental regulators and dairy farming subjects, it is necessary to mark the sign of each variable, in which: the cost of inspection by environmental regulators is *C*_1_; the cost of the dairy farming subject to participate in the green transformation of production is *C*_2_; dairy farming subjects who do not participate in the green transformation of production are fined for *G*; the bribes given to the environmental regulators through the rent-seeking of dairy farming subjects is *μC*_2_ (0 < *μ* < 1); the environmental regulators illegally charge penalties of *σG* (*σ* > 1) to the dairy farming subjects; the environmental regulator who gives the bribe and fine to the higher authority to obtain the incentive money is *τ*(*μC*_2_ + *G*) (0 < *τ* < *μ* < 1); the loss of government reputation when environmental regulators do not perform inspections and dairy farming subjects do not participate in the green transformation of production is *θ*_1_; the loss of government reputation when environmental regulators perform inspections but there is “regulation capture” is *θ*_2_; dairy farming subjects not involved in the green transformation of production inspected by environmental regulators need to bear the reputation loss of *η*_1_; and dairy farming subjects involved in the green transformation of production that are inspected by environmental regulators would obtain reputational gains of *η*_2_. Meanwhile, it is assumed that the probability of environmental regulators strictly enforcing the law and executing the green transformation of the production of dairy farming subjects is *p*_1_; the probability of no inspection shall be (1 − *p*_1_); the probability of participation of dairy farming subjects in the green transformation of production is *p*_2_; the probability of non-participation is (1 − *p*_2_); the probability that an environmental regulator obtains “regulation capture” is *p*_3_; the probability that it does not obtain “regulation capture” is (1 − *p*_3_); the rent-seeking probability of the dairy farming subject is *p*_4_; and the non-rent-seeking probability is (1 − *p*_4_). The symbols and meanings of the variables in the dynamic game between environmental regulators and dairy farming subjects is shown in Table 2.

Based on the above hypotheses, this paper constructs a dynamic game model between environmental regulators and dairy farming subjects to investigate the interaction mechanism between the environmental regulation and green transformation of dairy farming subject production, which includes the following seven scenarios (as shown in Figure 2):

Scenario 1: When environmental regulators perform strict inspections and dairy farming subjects participate in the green transformation of production, the dairy farming subject performs the green transformation at a cost of *C*_2_ and receives a reputational benefit of *η*_2_, whereas the inspection cost of *C*_1_ is borne by environmental regulators. At this point, the benefit of both parties is expressed as (−*C*_1_, −*C*_2_ + *η*_2_).

Scenario 2: When the environmental regulator performs a strict inspection and there is no “regulation capture”, and the dairy farming subject does not participate in the green transformation of production and does not seek rent from the environmental regulator, the dairy farming subject is fined *G*, and suffers reputational loss *η*_1_ for not participating in the green transformation of production, whereas the environmental regulators hand over the collected fine to the higher authority to obtain reward money *τG*. At this point, the benefit of both parties is expressed as (*τG* − *C*_1_, −*G* − *η*_1_).

Scenario 3: When the environmental regulator performs a strict inspection and there is no “regulation capture”, and the dairy farming subject does not participate in the green transformation of production but seeks rent from the environmental regulators, the bribe *μC*_2_ will be confiscated and a fine *G* will be imposed for not participating in the green transformation of production, and a reputation loss *η*_1_ will be incurred for not participating in the green transformation of production, whereas the environmental regulators will give the bribe and the fine to the higher authority to obtain the reward money *τ*(*μC*_2_ + *G*). At this point, the benefit of both parties is expressed as (*τ*(*μC*_2_ + *G*) − *C*_1_, −*μC*_2_ − *G* − *η*_1_).

Scenario 4: When the environmental regulator performs a strict inspection and there is “regulation capture”, and when the dairy farming subject does not participate in the green transformation of production and does not seek rent from the environmental regulator, the cost to be borne by the dairy farming subject includes the environmental regulator illegally collecting a fine *σG*, and at the same time, bears a reputation loss *η*_1_ for not participating in the green transformation of production, whereas the environmental regulator receives an illegally collected fine *σG*, but also needs to bear a reputation loss *θ*_2_. At this point, the benefit of both parties is expressed as (*σG* − *C*_1_ − *θ*_2_, −*σG* − *η*_1_).

Scenario 5: When the environmental regulator performs a strict inspection and there is “regulation capture”, and the dairy farming subject does not participate in the green transformation of production but seeks rent from the environmental regulator, the dairy farming subject needs to bear the costs including bribe *μC*_2_ and reputation loss *η*_1_, whereas the environmental regulator receives the bribe *μC*_2_, but also bears the reputation loss *θ*_2_. At this point, the benefit of both parties is expressed as (*μC*_2_ − *C*_1_ − *θ*_2_, −*μC*_2_ − *η*_1_).

Scenario 6: When environmental regulators do not perform strict inspections and dairy farming subjects participate in the green transformation of production, the cost of the green transformation of production is *C*_2_. In this case, the benefit of both parties is (0, −*C*_2_).

Scenario 7: When environmental regulators do not perform strict inspections and dairy farming subjects do not participate in the green transformation of production, environmental regulators are required to bear the reputational loss *θ*_1_. In this case, the benefit for both parties is (−*θ*_1_, 0).

Dynamic game scenarios between environmental regulators and dairy farming subjects, and the results of benefits for both, are shown in Table 3.

## 4. Results

### 4.1. Model Solving

To find the probabilistic equilibrium solution of the dynamic game model between environmental regulators and dairy farming subjects, this paper will adopt a bottom-up strategy through the method of reverse induction, starting from the bottom of the game tree. The first step is to seek the expected benefits of the environmental regulator and the dairy farming subject; then, the probabilistic equilibrium solutions for environmental regulators’ regulation capture or not, dairy farming subjects’ rent-seeking or not, environmental regulators’ inspection or not, and dairy farming subjects’ production green transformation participation or not would be obtained based on expected benefits.

First of all, from the bottom of the game tree, we examine the expected benefits of the environmental regulators in the case of regulation capture by the environmental regulators, and let the expected benefit be *ω*_1_, which is calculated as:(1)$$\begin{array}{c} \frac{\partial \omega_{1}}{p_{3}}=p_{4}\left(\mu C_{2}-C_{1}-\theta_{2}\right)+\left(1-p_{4}\right)\left(\sigma G-C_{1}-\theta_{2}\right)\\ \;\;-p_{4}\left[\tau \left(\mu C_{2}+G\right)-C_{1}\right]-\left(1-p_{4}\right)\left(\tau G-C_{1}\right)=0 \end{array}$$

To find the first-order partial derivative of Equation (1) and make the first-order partial derivative equal to 0, then we obtain:(2)$$\begin{array}{c} \frac{\partial \omega_{1}}{p_{3}}=p_{4}\left(\mu C_{2}-C_{1}-\theta_{2}\right)+\left(1-p_{4}\right)\left(\sigma G-C_{1}-\theta_{2}\right)\\ \;\;-p_{4}\left[\tau \left(\mu C_{2}+G\right)-C_{1}\right]-\left(1-p_{4}\right)\left(\tau G-C_{1}\right)=0 \end{array}$$

The equilibrium solution for the probability of rent-seeking by the environmental regulator in the case of regulation capture for the dairy farming subject is expressed as:(3)$$p_{4}^{*}=\frac{\left(\sigma -\tau \right)G-\theta_{2}}{\sigma G-\mu \left(1-\tau \right)C_{2}}$$

Then, from the bottom of the game tree, we examine the expected benefit of the dairy farming subject in the rent-seeking situation, and let the expected benefit be *ω*_2,_ which is calculated as:(4)$$\begin{array}{c} \omega_{2}=p_{3}p_{4}\left(-\mu C_{2}-\eta_{1}\right)+p_{3}\left(1-p_{4}\right)\left(-\sigma G-\eta_{1}\right)\\ +\left(1-p_{3}\right)p_{4}\left(-\mu C_{2}-G-\eta_{1}\right)+\left(1-p_{3}\right)\left(1-p_{4}\right)\left(-G-\eta_{1}\right) \end{array}$$

Finding the first-order partial derivative of Equation (4) and making the first-order partial derivative equal to 0, we obtain:(5)$$\begin{array}{c} \frac{\partial \omega_{2}}{p_{4}}=p_{3}\left(-\mu C_{2}-\eta_{1}\right)-p_{3}\left(-\sigma G-\eta_{1}\right)+\left(1-p_{3}\right)\left(-\mu C_{2}-G-\eta_{1}\right)\\ \;\;-\left(1-p_{3}\right)\left(-G-\eta_{1}\right)=0 \end{array}$$

The equilibrium solution for the probability of “regulation capture” of environmental regulators in a rent-seeking situation for the dairy farming subject is as follows:(6)$$p_{3}^{*}=\frac{\mu C_{2}}{\sigma G}$$

Thirdly, from the middle level of the game tree, we examine the expected benefit of the environmental regulator in the case of the environmental regulator performing an inspection, and let the expected benefit be *φ*_1_, which is calculated as:(7)$$\varphi_{1}=p_{1}p_{2}\left(-C_{1}\right)+p_{1}\left(1-p_{2}\right)\omega_{1}^{*}+\left(1-p_{1}\right)p_{2}\times 0+\left(1-p_{1}\right)\left(1-p_{2}\right)\left(-\theta_{1}\right)$$
where *ω*_1_^*^ in Equation (7) is obtained by bringing Equations (3) and (6) into Equation (1), as:(8)$$\omega_{1}^{*}=p_{4}^{*}\tau \left(\mu C_{2}+G\right)-C_{1}$$

To find the first-order partial derivative of Equation (7) and make the first-order partial derivative equal to 0, we obtain:(9)$$\frac{\partial \varphi_{1}}{p_{1}}=p_{2}\left(-C_{1}\right)+\left(1-p_{2}\right)\omega_{1}^{*}-\left(1-p_{2}\right)\left(-\theta_{1}\right)=0$$

The equilibrium solution for the probability of participation of the dairy farming subject in the green transformation of production under the enforcement inspection scenario by the environmental regulator was found to be:(10)$$p_{2}^{*}=1-\frac{C_{1}}{\omega_{1}^{*}+C_{1}+\theta_{1}}$$

Fourthly, the expected returns of the dairy farming subject in the case of the green transformation of production participation from the middle level of the game tree are examined, and the expected benefit is set to *φ*_2_, which is calculated as:(11)$$\begin{array}{c} \varphi_{2}=p_{1}p_{2}\left(-C_{1}+\eta_{2}\right)+p_{1}\left(1-p_{2}\right)\omega_{2}^{*}\\ \;+\left(1-p_{1}\right)p_{2}\left(-C_{2}\right)+\left(1-p_{1}\right)\left(1-p_{2}\right)\times 0 \end{array}$$

*ω*_2_^*^ is obtained by bringing Equations (3) and (6) into Equation (4), as:(12)$$\omega_{2}^{*}=p_{4}^{*}\left(-\mu C_{2}-G\right)-\eta_{1}$$

Finding the first-order partial derivative of Equation (11) and making the first-order partial derivative equal to 0, we obtain:(13)$$\frac{\partial \varphi_{2}}{p_{2}}=p_{1}\left(-C_{1}+\eta_{2}\right)-p_{2}\omega_{2}^{*}+\left(1-p_{1}\right)\left(-C_{2}\right)=0$$

The equilibrium solution for the probability of inspection by the environmental regulator in the case of the green transformation of production involving the dairy farming subject is found to be:
(14)$$p_{1}^{*}=\frac{C_{2}}{\eta_{2}}-\omega_{2}^{*}$$

### 4.2. Result Analysis

The probabilistic equilibrium solution of the dynamic game model between environmental regulators and dairy farming subjects calculated above is analyzed as follows:

According to Equation (14), which gives the equilibrium solution of the probability of inspection *p*_1_ in the case of the green transformation of dairy farming subject production, we could find that: First of all, the higher the green transformation of production cost (*C*_2_) for dairy farming subjects, the greater the probability that environmental regulators will perform inspections. Based on the premise of the “rational economic man”, whether dairy farming subjects can actively participate in the green transformation of production depends on the human cost, material cost, time cost, etc. When the cost is too expensive, dairy farming subjects will do their best to disguise their polluting behavior, whereas the environmental regulators will then urge them to increase their willingness to protect the environment and force them to participate in the green transformation of production through meticulous planning. Secondly, the higher the reputational gain (*η*_2_) obtained by the dairy farming subject in the process of participating in the green transformation of production, the more active, proactive, and dynamic their participation in the green transformation of production will be, and the more they will be able to consciously protect the ecology and sustain it, at which point, the pressure of inspection by the environmental regulators will become less. Thirdly, the higher the expected benefit (*ω*_2_^*^) of the dairy farming subject in the first stage of the game, the greater the expected reputational loss it experiences after being inspected by the environmental regulator. In order to minimize losses and maximize benefits, the dairy farming subject is more capable to proactively participate in the green transformation of production, at which point, the environmental regulators will naturally reduce the probability of inspection due to the consideration of cost minimization.

Furthermore, according to Equation (10), the equilibrium solution of the probability of green transformation of dairy farming subject production *p*_2_ in the case of enforcement inspection by the environmental regulator can be found as follows: Firstly, the higher the inspection cost (*C*_1_) of the environmental protection regulator, the smaller the probability that it will perform an inspection on the dairy farming subject; when the probability that the dairy farming subject will be found and punished for not participating in the green transformation of production is reduced, the easier it will be to have violations, and the probability of participating in the green transformation of production will be significantly reduced in this case. Secondly, the greater the loss to the government’s reputation (*θ*_1_) due to the environmental regulator’s failure to perform inspections, the greater the tension in the principal–agent relationship between the government and the environmental regulator, which forces the environmental regulator to increase the punishment for the dairy farming subject’s pollution behavior and to increase the frequency of urging the dairy farming subject to participate in the green transformation of production; thus, the probability of the dairy farming subject’s green transformation of production is then significantly increased. Thirdly, the higher the expected benefit (*ω*_1_^*^) of the environmental regulators in the first stage of the game, the higher the probability of the green transition in the production of the dairy farming subject. Essentially, the expected benefits of environmental regulators are a function of the cost of violations and reputation loss of dairy farming subjects, and the higher the cost of violations and the greater the reputation loss would be, the more likely dairy farming subjects can actively engage in the green transformation of production.

In addition, the equilibrium solution of Equation (6) for the probability *p*_3_ of “regulation capture” of the environmental regulator in the rent-seeking scenario for dairy farmers shows: First of all, the more bribes dairy farming subjects give for rent-seeking from environmental regulators (*μC*_2_), the more they are motivated to take bribes from environmental regulators, and the greater the probability is that they will obtain “regulation capture”. Secondly, the dairy farming subject will weigh the costs of illegal fines collected by the environmental regulator, the costs of the green transformation of production, and the reputational benefits. If the environmental regulator increases the penalty (*σ*) for the dairy farmers’ pollution behavior, the high penalty (*σG*) will force the dairy farming subject to transform, and this transformation will reduce the probability of rent-seeking by the dairy farmers; therefore, the probability of the environmental regulator obtaining “regulation capture” will be reduced. This also reflects a real problem: if environmental regulators want to abuse their powers and accept rent-seeking, it is necessary to improve the punishment to restrain the dairy farming subjects, but this punishment also needs to be “perfectly appropriate” as the amount of the fine should be controlled below the cost of the green transformation of production cost while also meeting the need to form environmental pressure on the dairy farming subject.

Fourthly, from Equation (3), the equilibrium solution of the rent-seeking probability *p*_4_ of dairy farming subjects in the case of regulation capture by environmental regulators shows: First of all, when the gap between the bribe money (*μC*_2_) and the government’s reward money (*μτC*_2_) to the environmental regulator is larger, the easier it is for the environmental regulator to be captured by the dairy farming subject, causing it to join with the dairy farming subject to deceive the government. Once this illegal association exists for a long time, it will not only bring serious negative impacts to the ecological environment, but will also seriously damage the credibility of the government, and the probability of rent-seeking for the dairy farming subject in the process will be greatly enhanced. Secondly, when the environmental regulator’s violations bring negative impacts to the government’s credibility, it will certainly be severely criticized and educated by the government; serious cases may face criminal penalties, and thus, when the environmental protection regulator is under pressure from the government’s disciplinary action, the less feasible it is that it will be captured by the dairy farming subject, and the probability of rent-seeking will naturally be reduced. Thirdly, the amount of rent-seeking by dairy farming subjects and the probability of environmental regulators being captured are positively correlated with each other, but since the government, in its capacity as a principal, exerts pressure on environmental regulators, it must be able to restrain environmental regulators from operating in violation of the law and also inhibit rent-seeking by dairy farming subjects.

## 5. Discussion

A comprehensive analysis of the construction and results of the above-mentioned dynamic game model between environmental regulators and dairy farming subjects shows that environmental pollution of dairy farming cannot be underestimated, and environmental regulation plays a crucial role in this process. However, the “invisible hand” of the market has led to the existence of the risk of rent-seeking by dairy farming subjects and to “regulation capture” of environmental regulators, which also illustrates from another perspective that there are certain defects and dilemmas in the process of environmental regulation in promoting the green transformation of dairy farming subject production. Specifically, some limitations are listed as follows:

First of all, environmental regulation policies are “introduced but not implemented”, increasing the risk of rent-seeking by dairy farming subjects [47]. It is undeniable that the Chinese government has made great efforts in promoting the green transformation of dairy farming subject production. Especially since the 18th Party Congress, according to incomplete statistics, the National People’s Congress, the State Council, the General Office of the State Council, and national ministries and commissions have introduced nearly more than 30 laws, regulations, and policy documents related to the disposal and resource utilization of livestock and poultry waste. The introduction of these policies certainly laid out important guidance to solve the problem of environmental pollution in dairy farming. These problems are united as the government, in the process of promoting the green transformation of dairy farming, “want to be difficult as” possible, specifically in the regulation policy “more introduction, less implementation” as the green transformation goals do not match the reality of constraints. For example, the “livestock and poultry scale farming pollution prevention and control regulations” mentioned the organic fertilizer production tax incentives, transport preferences, the standard of fertilizer subsidies, biogas power generation to enjoy preferential feed-in tariffs, and new energy preferential policies. However, no nationwide policy implementation rules and standards have been made thus far. In combination, environmental regulation policies ignore the protection of dairy farming subjects, and instead, increase their risk of rent-seeking.

Secondly, the subject of environmental regulation, “decentralization, less inspection”, increases the risk of environmental protection regulators obtaining “regulation capture”. The “environmental decentralization” refers to the gradual decentralization of the environmental governance authority and responsibility moving from the central government to local governments. Local governments are responsible for the implementation and enforcement of the central environmental policy and are subject to the inspection of the central government. Theoretically, local governments are given a certain amount of discretion in environmental matters and are able to tailor their policies to regional developments. However, limiting policies to the dairy farming industry means that they not only cannot bring in financial income for the local government, but also they need to spend a lot of human, material, and financial resources on the main production of the dairy farming green transformation behavior to perform inspections. Therefore, the local government’s division of responsibilities for environmental regulation appears to be cascading down, with most practical work falling to the grassroot village and town level, making grassroots work stressful and costly to regulate, which can easily lead to collusion between grassroot environmental regulators and dairy farming subjects, working together to deceive higher authorities and losing the binding power of environmental regulation itself.

Thirdly, the environmental regulation approach, “results-oriented, light on the process”, reduces the enthusiasm of the dairy farming subjects to be involved in the green transformation of production. The seriousness of environmental pollution in dairy farming forced the government to give administrative orders to take control, but most of this control is to order rectification, discipline, and to shut down, which is mainly concerned about the “treatment”, but there is a lack of awareness of source reduction and the process control of “prevention”. For example, in accordance with the “livestock and poultry scale farming pollution prevention and control regulations” and “water pollution action plan”, the requirements from 2017 were previously around the delineation of the ban area and were introduced one after another, with the ban focused on large-scale farming in the area. However, the national level is not limited to the “scale”, and the right to delegate has been given to the local government; therefore, there is a commanding means of “one size fits all” and “no breeding areas” has become a “no livestock area”. The long-term development of the dairy farming industry cannot be separated from the constraints of environmental regulations, but these constraints must not harm the benefits of the main breeding, must not compete with the political tasks as the goal, and must not be at the expense of the breeding subject. We need to combine “cure” and “prevention”, taking into account the process and results of the industry.

Fourthly, the form of environmental regulation emphasizes “formal regulation, but not informal regulation”, reducing the effect of environmental regulation to “forced transformation”. Environmental regulation encompasses not only formal environmental regulation, but also informal environmental regulation. Under the current Chinese economic system, the former is used frequently. The earliest studies on the environmental regulation of livestock and poultry farming drew conclusions from industry and manufacturing, again with a focus on formal environmental regulation. Compared with the growth cycle curve of industrial and manufacturing industries, the livestock and poultry farming industry, especially the dairy farming industry, has a unique vulnerability and a long growth cycle. Formal environmental regulation can indeed solve the environmental pollution problem of dairy farming in the short term, but it also invariably hinders the development of the industry and increases the pressure on the industry. In addition, compared with the decision-making ability of entrepreneurs in industry and manufacturing, dairy farming subjects are more rooted in rural areas and have grown up as farmers who have been engaged in agricultural production for a long time, and their behavioral decisions are easier to guide and improve. Under the inspection and control of formal environmental regulation, the potential of informal environmental regulation can be properly developed to accelerate the green transformation of dairy farming subject production.

## 6. Conclusions 

The following conclusions were drawn from the study: (1) The green transformation of dairy farming subject production follows the opinion of “subject substitution”, emphasizing “emission reduction” as well as “efficiency enhancement”. (2) Strictly designed environmental regulations are sufficient to drive the green transformation of production by dairy farming subjects, but inevitably, environmental regulators are vulnerable to rent-seeking behavior and “regulation capture”. (3) The introduction of the reputation mechanism has greatly reduced the rent-seeking behavior of dairy farming subjects and the probability of environmental regulators being “captured”, indirectly forcing dairy farming subjects to participate in the green transformation of production. (4) The greater the penalty for non-participation of dairy farming subjects in the green transformation of production, the more it can force them to participate in the green transformation of production, while also reducing the risk of damage to the credibility of the government. (5) Further discussing the shortcomings and dilemmas of environmental regulation, we found that under the current system, environmental regulation policies “emphasize introduction but not implementation”, environmental regulation subjects “emphasize decentralization but not regulation”, environmental regulation methods “emphasize results but not processes”, and environmental regulation forms “emphasize formal regulation but not informal regulation”.

The following policy implications are proposed for the shortcomings of environmental regulation itself: First of all, the environmental regulation policy should be “introduced” and “implemented” to precisely control the time node; to achieve an effective interface between policy and policy, not “introduced for the sake of introduction”; to drive the main production of the green transformation of farming for classification, zoning, and point implementation; and to protect the interests of the main farming and reduce the risk of rent-seeking. Secondly, the subject of environmental regulation should both “focus on power sharing” and “focus on regulation”. Decentralization is meant to tailor policies to local conditions; therefore, the central government must do a good job of inspection while simplifying and decentralizing government to avoid causing collusion between grassroot environmental regulators and dairy farming subjects, as deception and falsification at every level not only does not effectively protect the ecological environment, but also derives from the dual cost of the loss of government credibility. Thirdly, environmental regulation should be both “result-oriented” and “process-oriented”. The result of ecological safety of dairy farming is important, but the result is based on the greening of the whole production process. “Source reduction, process control, terminal treatment” is the essence of the green transformation of dairy farming’s main production, but also the essence of the path should be a specific operation. In addition, the form of environmental regulation must be both “formal and informal”. The remarkable effectiveness of formal environmental regulation already speaks for itself. In theory, informal environmental regulations can also arouse the environmental awareness of dairy farming subjects and motivate them to participate in the green transformation of production; thus, it is reasonable to pay attention to informal environmental regulations, and it is necessary to conduct further theoretical and empirical studies on the effect of informal environmental regulations in driving the green transformation of dairy farming subject production.

In conclusion, we have given the scientific connotation of the green transformation of dairy farming subject production through the research mentioned above, we have also conducted theoretical and empirical studies on the interaction mechanism between the environmental regulation and green transformation of dairy farming subject production, and we have discussed systematically the problems of environmental regulation itself. However, there are still two limitations of the study, which are important directions for further exploration in this area in the future: First, research should be conducted to obtain first-hand data through field visits, and to review and assess the development level of the green transformation of dairy farming subject production in order to test the scientificalness of the theoretical connotation of dairy farming subjects. Second, we need to use fieldwork data to empirically test the impact of environmental regulations on the green transformation of dairy farming subject production in order to test the reality of the interaction mechanism between environmental regulations and the green transformation of dairy farming subject production.

## Figures and Tables

**Figure 1 ijerph-19-12982-f001:**
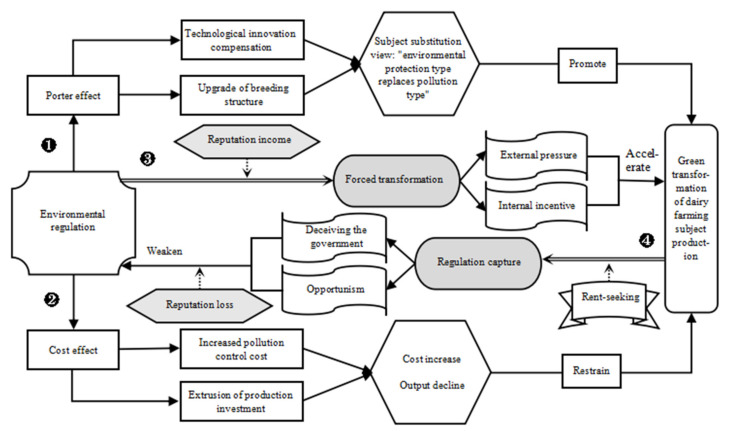
Interaction mechanism between environmental regulation and green transformation of dairy farming subject production.

**Figure 2 ijerph-19-12982-f002:**
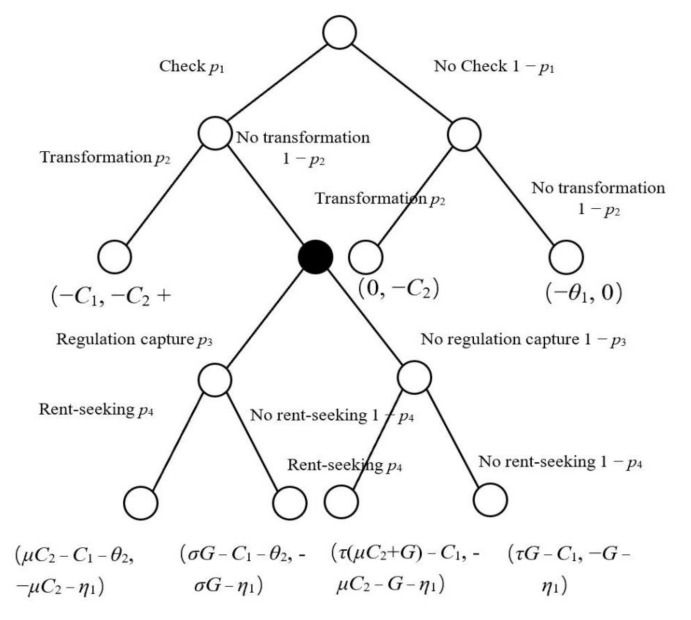
Dynamic game model between environmental regulators and dairy farming subjects.

**Table 1 ijerph-19-12982-t001:** Evaluation index system for green transformation of dairy farming subject production.

First-Level Indicators	Second-Level Indicators	Third-Level Indicators	Computing Method	Nature
Emission reduction	Non-point source pollution reduction	Chemical oxygen demand	Total chemical oxygen demand emissions (tons)	-
		Nitrogen	Total nitrogen emissions (tons)	-
		Phosphorus	Total phosphorus emissions (tons)	-
		Zinc	Total zinc emissions (tons)	-
		Copper	Total copper emissions (tons)	-
	Carbon emission reduction	Carbon emission	Total carbon dioxide emissions (tons)	-
Efficiency enhancement	Benefit increase	Land cost	Land cost per unit of dairy farming (RMB/head)	-
		Labor cost	Labor cost per unit of dairy farming (RMB/head)	-
		Material and service cost	Material and service cost per unit of dairy farming (RMB/head)	-
		Milk product output	Milk product output per unit of dairy farming (million RMB/head)	+
		By-product output	By-product output per unit of dairy farming (million RMB/head)	+
	Efficiency increase	Dairy farming efficiency	Measurement based on ultra-efficient SBM model	+

**Table 2 ijerph-19-12982-t002:** The symbols and meanings of the variables in the dynamic game between environmental regulators and dairy farming subjects.

Environmental Regulators		Dairy Farming Subject	
Content	Symbol	Content	Symbol
Inspection cost	*C* _1_	Cost for green transformation	*C* _2_
Penalties charged illegally	*σG*(*σ* > 1)	Fine for not being involved in green transformation	*G*
Incentive money from higher authority	*τ*(*μC*_2_ + *G*)(0 < *τ* < *μ* < 1)	Bribes for rent-seeking	*μC* _2_
Reputation loss for not performing inspections	*θ* _1_	Reputation loss for not being involved in green transformation and inspection	*η* _1_
Reputation loss for regulation capture in inspections	*θ* _2_	Reputation gained for being involved in green transformation and inspection	*η* _2_
Inspection (no inspection)	*p*_1_(1 − *p*_1_)	Transformation (no transformation)	*p*_2_(1 − *p*_2_)
Regulation capture (no regulation capture)	*p*_3_(1 − *p*_3_)	Rent-seeking (no rent-seeking)	*p*_4_(1 − *p*_4_)

**Table 3 ijerph-19-12982-t003:** Dynamic game scenarios between environmental regulators and dairy farmers and results of benefits for both.

Type		Dairy Farming Subject						
		△	▽+◁	▽+▷	▽+◁	▽+▷	△	▽
**Environmental regulators**	▲	(−*C*_1_, −*C*_2_ + *η*_2_)						
	▲+◀		(*τG* − *C*_1_, −*G* − *η*_1_)					
	▲+◀			(*τ*(*μC*_2_ + *G*) − *C*_1_, −*μC*_2_ − *G* − *η*_1_)				
	▲+▶				(*σG* − *C*_1_ − *θ*_2_, −*σG* − *η*_1_)			
	▲+▶					(*μC*_2_ − *C*_1_ − *θ*_2_,−*μC*_2_ − *η*_1_)		
	▼						(0, −*C*_2_)	
	▼							(−*θ*_1_, 0)

Note: In the case of environmental regulators, “▲” is used for “inspection” and “▼” is used for “no inspection”; “▶” is used for “regulation capture” and “◀” is used for “no regulation capture”. In terms of the dairy farming subject, “involved in the green transformation of production” is indicated by “△” and “not involved” is indicated by “▽”; “rent-seeking” is indicated by “▷” and “no rent-seeking” is indicated by “◁”.

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
