# Peer review of "“Forced Transformation” or “Regulation Capture”—Research on the Interactive Mechanism between Environmental Regulation and Green Transformation of Dairy Farming Subject Production"

_ijerph, 2022, doi:10.3390/ijerph191912982_

Round 1

Reviewer 1 Report

The aim of this paper is to investigate the interactive mechanism between environmental regulation and green transformation of dairy farming main production, build a dynamic game model between the environmental protection supervision department and dairy farming subjects, and introduce the constraints and benefits of reputation mechanism on their behavior into the model to explore whether environmental regulation can drive the green transformation of the dairy farming main production.

My opinions and suggestions regarding the study are given below;

Most past studies have focused on the environmental problems of pig farming and have also ignored dairy farming, which has become a "major pollutant". In this study, analyzes were made considering dairy farming. I found the subject of the study interesting and original. In the study, literature review has been done sufficiently. The findings of the study were discussed in detail and policy recommendations were presented based on the findings. In the study, both the necessary theoretical background is explained and the subject is analyzed using the game theoretical model.

The paper is generally quite well written. I found the study successful. But I suggest some minor fixes.

First of all, it would be appropriate to clearly state the contribution of the study to the literature in the introduction. In this way, the reader can more easily understand the difference of the study from the previous studies.

My second suggestion is about the limitations of the study. The limitations of the study are not specified in the article. It would be appropriate to state the limitations of the study in the conclusion section. In addition, what recommendations are made for future studies.

Reviewer 2 Report

The author should revise the following questions.

Reviewer 1: "Forced Transformation" or "Regulation Capture"—Research on the Interactive Mechanism between Environmental Regulation and Green Transformation of Dairy Farming Subject Production: ijerph-1924713.

First, The Reviewer would like to thank the editors of " International Journal of Environmental Research and Public Health" for sending me the manuscript for review. The reviewers also cherish this opportunity to review the manuscript and will carefully review the manuscript.

The author's research ideas and methods are clear. There are innovative research applications and presentations that are not sufficiently clear, and There were many problems in the content analysis of the manuscript. There are many unnecessary descriptions in the text, which are too cumbersome to express the research. The comprehensive judgment is the major revision. The current format does not meet the requirements of SCI and journals, and the author is requested to revise it carefully according to the reviewers' comments.

The author should revise the following questions:

1. Page 1, 13-45; Please revise the abstract mainly to explain several issues: existing problems, research methods, research conclusions, research innovation, and the value of achieving results. Please revise the keywords, do not duplicate the title, and affect the search scope of your manuscript.

2. Page 2, 61-63; ①②Modify into citation format and insert.

3. Page 2, 47-97;Please revise this section; what conclusions did the author draw from reading the literature, and does the literature fully prove the authenticity of your research question and the important value of your research? The question raised lacks an analysis of the literature. Add the questions to be solved by the research in subsections in which aspects of the innovation of the manuscript are presented, and briefly explain how to demonstrate the research innovation.

4. Page 3-4, 99-187; A literature review aims to demonstrate the importance and value of your research question through published results and to illustrate the current state of research in the field. The content of literature unrelated to your research in the manuscript is not required, and each research question cites 2-3 recent representative literature analyses. Please revise section 2.

5. Page 4, 193-212; Please put the content of this paragraph at the end of the introduction, briefly explain the core content of the research, and do not need to describe more.

6. Page 5-7, 213-334; Please reduce the content of the relevant research process. After analyzing the influencing factors, the weight of the influencing factors should be considered. The degree of each influencing factor to the core content of the research is measured by indicators. There is a coupling effect.

7. Page 8, 371-410; The manuscript aims to analyze the influencing factors of government regulatory agencies. The manuscript is divided into two situations of management agency action and inaction, and some conclusions are drawn. Please add references and relevant legal and regulatory documents to improve the scientific and rigor of your analysis process. This research method is imprecise in obtaining key conclusions through simple language descriptions.

8. Page 5,9, 213,423; Please revise the title to refer to the SCI scientific literature format.

9. Page 11-12, 551-576; Please make a Table to put the data in.

10. Page 12-13, 584-624; Please make a Table to put the data in. Add references for some hypothetical models.

11. Page17, 809-810;Please add a citation.

12. Page16-18, 782-883; The author is asked to simplify the sixth section, and the core reasons, theoretical model system and process of the analysis can be clearly described.

13. Page19-20, 885-937; Please shorten the repetitive narrative part; whether there are new research conclusions and content that needs to be highlighted and explained needs to be added; supplement the research flaws and deficiencies of the manuscript and the fields and directions that need to be expanded in the future.

14. Page1-20, 13-951; Ask professional translators from native-speaking countries to proofread the manuscript. The original manuscript contains many grammatical and wording errors, and the expression of sentences is irregular and unclear.

Original manuscript: Under the situation of increasing resource and environment shortage, the green transformation of dairy farming production driven by environmental regulation is the concentrated embodiment of "promising government" to solve the problem of breeding environmental pollution. Due to the shortcomings of environmental regulation itself and the undefined connotation of the green transformation of the main production of dairy farming, the interactive relationship between the two remains to be unclear at present. Based on defining the concept of green transformation of dairy farming main production, this paper aims to analyze the interactive mechanism between environmental regulation and green transformation of dairy farming main production, build a dynamic game model between the environmental protection supervision department and dairy farming subjects, and introduce the constraints and benefits of reputation mechanism on their behavior into the model to explore whether environmental regulation can drive the green transformation of the dairy farming main production. The results showed that the green transformation of dairy farming main production followed the "main substitution view", emphasized "source reduction, process control and end use"; strictly designed environmental regulation could effectively drive the production green transformation of dairy farming subjects, but it was inevitable that the environmental protection supervision department was vulnerable to the rent-seeking behavior of dairy farming subjects, which was "captured by regulation"; the introduction of reputation mechanism has greatly improved the rent-seeking behavior of dairy farming subjects and the probability that environmental protection regulators are "captured by regulation", indirectly forcing dairy farming subjects to participate in the green transformation of production; the greater the punishment for dairy farming subjects who do not participate in the green transformation of production was, the more they can be forced to participate in the green transformation of production was. At the same time, it also reduces the risk of damage to the credibility of the government. Based on the above studies, this paper also further discussed the shortcomings of environmental regulation itself, including "re exit and light implementation" of environmental regulation policy, "decentralization and light supervision" of environmental regulation subject, "result and light process" of environmental regulation mode, and "formal regulation and light informal regulation" of environmental regulation form,
which provides a scientific reference for the formulation of the environmental regulation policy of livestock and poultry breeding in the future.

After modification:
Under the increasing resource and environmental shortage, the green transformation of dairy farming production driven by environmental regulation is the concentrated embodiment of a "promising government" to solve the problem of breeding environmental pollution due to the shortcomings of environmental regulation itself and the undefined connotation of the green transformation of the primary production of dairy farming, the interactive relationship between the two remains to be unclear at present. Based on defining the concept of green transformation of dairy farming's primary production. This paper analyzes the interactive mechanism between environmental regulation and the green transformation of dairy farming primary production. Build a dynamic game model between the environmental protection supervision department and dairy farming subjects and introduce the constraints and benefits of reputation mechanism on their behavior into the model to explore whether environmental regulation can drive the green transformation of the dairy farming primary production. The results showed that the green transformation of dairy farming primary production followed the "main substitution view", emphasized "source reduction, process control and end use"; strictly designed environmental regulation could effectively drive the production green transformation of dairy farming subjects, but it was inevitable that the environmental protection supervision department was vulnerable to the rent-seeking behavior of dairy farming subjects, which was "captured by regulation"; the introduction of reputation mechanism has dramatically improved the rent-seeking behavior of dairy farming subjects and the probability that environmental protection regulators are "captured by regulation", indirectly forcing dairy farming subjects to participate in the green transformation of production; the more significant the punishment for dairy farming subjects who do not participate in the green transformation of production was, the more they can be forced to participate in the green transformation of production was. At the same time, it also reduces the risk of damage to the government's credibility. Based on the above studies, this paper also further discussed the shortcomings of environmental regulation itself. Including "re exit and light implementation" of environmental regulation policy, "decentralization and light supervision" of the environmental regulation subject. "Result and light process" of environmental regulation mode, and "formal regulation and light informal regulation" of environmental regulation form, which provides a scientific reference for the formulation of the environmental regulation policy of livestock and poultry breeding in the future.

15. Page21-22, 952-1030;Please check the references, some references cannot be searched.

Round 2

Reviewer 2 Report

The author should revise the following questions:

Reviewer 1: "Forced Transformation" or "Regulation Capture"—Research on the Interactive Mechanism between Environmental Regulation and Green Transformation of Dairy Farming Subject Production: ijerph-1924713.

The manuscript has been greatly improved after a lot of revisions, and there are still some details that need to be carefully checked and revised by the author. The comprehensive judgment is the minor revision. The current format does not meet the requirements for journal publication.

The author should revise the following questions:
1. Page 6, 280; Modify the format of the font in the Table (the first line of letters is bold), modify it according to the requirements of the journal, check the full text and modify it.

2. Page 7, 312; Please revise Fig.1. Mark and revise the full text according to the format requirements of the journal.

3. Page 14-15,589-645; Place the number of the formula to the right of the formula and modify the full text.

4. Page 1-15,208-729; Modify the titles 2, 3, 4, and 5 as standard SCI titles: 2: Methods; 3: Results; 4: Discussion. 5: Conclusion. Author, you did not find that the theoretical model system, data results and data analysis of the manuscript are not clearly divided .The lines are not clear, making it difficult for readers to read and understand.

5. Page 1-20,16-975; Authors need to carefully check the format defects of the manuscript according to the format requirements of the journal and revise all the manuscripts to the standard publication format. For example: italic marks for special symbols; incorrect placement of numbers in citations (7;30); capitalization of letters and symbols; punctuation marks (Table 1 xxxxxxxxx.) and so on.

Author Response

Dear reviewer,

Thank you for your valuable feedback of our manuscript entitled "Forced Transformation" or "Regulation Capture"—Research on the Interactive Mechanism between Environmental Regulation and Green Transformation of Dairy Farming Subject Production. Those comments are all valuable and very helpful for revising and improving our paper, as well as the important guiding significance to our research.

We have carefully studied the comments of reviewer, and based on the comments, the paper has been modified as follows (the black font below is the comments of reviewer, and the red font below is the modification description):

Responses to the comments

Point 1: Page 6, 280; Modify the format of the font in the Table (the first line of letters is bold), modify it according to the requirements of the journal, check the full text and modify it.

Response 1: Thank you sincerely for this comment. We have revised all the tables in the article according to your latest comments to meet the journal publication standards. Please refer to the revised manuscript for details of the revised content.

Point 2: Page 7, 312; Please revise Fig.1. Mark and revise the full text according to the format requirements of the journal.

Response 2: Thank you sincerely for this comment. We have revised all the figures in the article according to your latest comments to meet the journal publication standards. Please refer to the revised manuscript for details of the revised content. Please refer to the revised manuscript for details of the revised content.

Point 3: Page 14-15,589-645; Place the number of the formula to the right of the formula and modify the full text.

Response 3: Thank you sincerely for this comment. We have modified the formula number in the article according to your latest comments, and put the formula number on the right of the formula. Please refer to the revised manuscript for details of the revised content.

Point 4: Page 1-15,208-729; Modify the titles 2, 3, 4, and 5 as standard SCI titles: 2: Methods; 3: Results; 4: Discussion. 5: Conclusion. Author, you did not find that the theoretical model system, data results and data analysis of the manuscript are not clearly divided .The lines are not clear, making it difficult for readers to read and understand.

Response 4: Thank you sincerely for this comment. We have revised all the titles of the article to make the article clearer for readers to read and understand. Please refer to the revised manuscript for details of the revised content.

Point 5: Page 1-20,16-975; Authors need to carefully check the format defects of the manuscript according to the format requirements of the journal and revise all the manuscripts to the standard publication format. For example: italic marks for special symbols; incorrect placement of numbers in citations (7;30); capitalization of letters and symbols; punctuation marks (Table 1 xxxxxxxxx.) and so on.

Response 5: Thank you sincerely for this comment. According to the format requirements of the journal, we have carefully checked the format defects of the manuscript and revised it to meet the standard of publication of the journal. Please refer to the revised manuscript for details of the revised content. Please refer to the revised manuscript for details of the revised content.